# Prevalence of Hypertensive Disorders, Antihypertensive Therapy and Pregnancy Outcomes among Pregnant Women: A Retrospective Review of Cases at Tamale Teaching Hospital, Ghana

**DOI:** 10.3390/ijerph20126153

**Published:** 2023-06-16

**Authors:** Amos Adapalala Bugri, Solomon Kwabena Gumanga, Peter Yamoah, Ebenezer Kwabena Frimpong, Manimbulu Nlooto

**Affiliations:** 1Tamale Teaching Hospital, Tamale P.O. Box TL16, Ghana; adapalala@yahoo.com (A.A.B.); gumangask@yahoo.co.uk (S.K.G.); 2School of Pharmacy, University of Health and Allied Sciences, Ho PMB 31, Ghana; 3Indigenous Knowledge Systems Centre, Faculty of Natural and Agricultural Sciences, North-West University, Private Bag X 2046, Mmabatho 2790, South Africa; pharmeben@gmail.com; 4Department of Pharmacy, University of Limpopo, Private Bag X 1106, Sovenga, Polokwane 0727, South Africa

**Keywords:** hypertensive disorders in pregnancy, eclampsia, pre-eclampsia, chronic kidney disease, perinatal morbidity and mortality

## Abstract

Hypertensive disorders associated with pregnancy are a major health concern and a leading cause of maternal indisposition and transience. The main objective of this study was to assess the prevalence of hypertension in pregnancy as well as antihypertensive therapy and pregnancy outcomes among pregnant women at Tamale Teaching Hospital (TTH) in Ghana. This was a retrospective study conducted using data from the folders of pregnant hypertensive patients. The study was conducted at the maternity ward of TTH from 1 June 2018 to 31 May 2019. Participants were all pregnant women with a diagnosis of hypertensive disorders. The prevalence of hypertensive disorders in pregnancy was 12.5%. The most common antihypertensive medication prescribed was sustained-release oral nifedipine, which was prescribed for 548 (81.4%) participants either alone or with methyldopa, followed by oral methyldopa: 506 (75.2%), intravenous hydralazine: 94 (14.0%), intravenous labetalol: 28 (4.2%) and diuretics: 10 (1.5%). Thirty-eight (5.7%) babies died before delivery, whereas 635 (94.3%) babies were born alive. Twenty-six out of the 38 dead babies (68.4%) were babies of pregnant women with elevated BP, whereas 12 (31.6%) were babies of those with normal BP. There was a statistically significant association between BP control and delivery outcomes. The study observed adherence to antihypertensive medicines recommended by the standard treatment guidelines of Ghana for the management of hypertensive disorders in pregnancy. The BPs of about two-thirds of the study participants were well controlled with the antihypertensive therapy. The majority of the study participants with well-controlled BP had positive delivery outcomes.

## 1. Introduction

Hypertension continues to be a global health lifestyle disease of major concern due to its high prevalence and associated cardiovascular and chronic kidney disease [1]. Hypertensive disorders in pregnancy have particularly become a public health challenge in many countries as they contribute significantly to maternal mortality and morbidity [2,3]. Globally, hypertensive disorders in pregnancy complicate about 6–8% of all pregnancies [4]. Most international guidelines on the management of hypertension have defined hypertension as having a clinical blood pressure (CBP) greater than or equal to 140/90 mmHg [5,6,7]. Hypertensive disorders in pregnancy can be grouped into four categories: chronic hypertension, pre-eclampsia/eclampsia, pre-eclampsia superimposed on chronic hypertension and gestational hypertension [8]. A systolic BP of at least 140 mmHg or a diastolic BP of at least 90 mmHg diastolic before pregnancy or before 20 weeks of gestation is an indication of chronic hypertension [9]. Pre-eclampsia is characterised by new onset hypertension and proteinuria in pregnancy and usually occurs after 20 weeks of gestation and can be superimposed on another hypertension disorder of pregnancy [10,11]. Chronic hypertension with superimposed pre-eclampsia is a situation in which women with chronic hypertension before pregnancy develop pre-eclampsia in addition to after 20 weeks of gestation [12]. Gestational hypertension is characterised by elevated blood pressure (systolic ≥ 140 or diastolic ≥ 90 mm Hg), no proteinuria and no manifestation of pre-eclampsia/eclampsia for pregnant women whose blood pressures were normal before pregnancy [13].

It is important to diagnose and treat hypertension in pregnancy very early to prevent adverse events in both the pregnant woman and her baby, which include increased risk of maternal stroke, lower birth weight and increased risk of the baby requiring neonatal intensive care. Knowledge of risk factors of hypertension in pregnancy as well as evidence-based pharmacotherapy based on national and international guidelines are essential in reducing the incidence of hypertension in pregnancy as well as improving its outcomes. In a study conducted in Ethiopia on risk factors associated with hypertensive disorders in pregnancy, it was observed that the major risk factors associated with hypertension in pregnancy were a maternal age of 35 years or more, rural residential dwelling, prim gravida, null parity, positive history of abortion, twin pregnancy, lack of antenatal care follow-up, pre-existing hypertension, family history of hypertension and a history of diabetes mellitus [12]. In a study on the distribution of maternal mortality among different socio-demographic groups in Ghana, it was observed that haemorrhage and infectious diseases such as malaria and viral hepatitis were implicated in most maternal deaths [13]. Furthermore, hypertensive disorders in pregnancy have been identified as the leading cause of maternal deaths in major teaching hospitals in Ghana [3,14]. This may not always indicate a poor level of care in the teaching hospitals but rather the fact that most referred cases from the peripheral facilities to the teaching hospitals are fatal, resulting in poor outcomes. Additionally, the first-contact healthcare teams in rural settings may not have adequate professional knowledge and skills to manage such cases. Ineffective treatment of pre-eclampsia leads to eclampsia. Eclampsia can occur before, during and after labour. Women with chronic hypertension intending to be pregnant must attend the clinic for medical care to reduce their high BP. Pregnant women with high BP must be treated with antihypertensive medicine, bearing in mind their benefits and risks to the mother and unborn baby. A study conducted at the Korle-Bu Teaching Hospital of Ghana showed that there was a significant burden of perinatal morbidity and mortality associated with hypertension-related disorders in the obstetric population of Ghana. The study further established that the adverse outcomes associated with pregnancy-related hypertension were more prevalent with pre-eclampsia [15]. Therefore, it is necessary that healthcare professionals assiduously screen and treat pregnant women with hypertensive disorders during antenatal visits.

Tamale Teaching Hospital (TTH), where this study was conducted, serves as a referral centre in the northern belt of Ghana. About 7000–8000 pregnant women attend TTH annually. The maternal mortality ratio was 890 per 100,000 live births in 2016 with hypertensive disorders being one of the main causes of death annually [16]. However, there is a paucity of data on hypertensive disorders in pregnancy, their management and treatment outcomes in the northern belt of Ghana. Therefore, the objective of this study was to assess the prevalence of hypertension in pregnancy as well as the antihypertensive therapy and pregnancy outcomes among pregnant women at TTH, which serves as a major referral facility in the northern belt of Ghana. It must be emphasised that the study did not evaluate the antihypertensive drugs used but demonstrated that blood pressure control is associated with better perinatal outcomes. Findings from this study are intended to add to the body of knowledge of the prevalence of hypertensive disorders, types of antihypertensive medicines prescribed, the extent of BP control with the medicines as well as pregnancy outcomes among pregnant women with elevated BP in Ghana and other low- and middle-income countries.

## 2. Materials and Methods

The study was retrospective in design, making use of data from patient folders, Health and Administration Management Software (Hospital Availability Exchange Version 2.0) and antenatal records of pregnant women [17]. The study was conducted at the maternity ward of TTH from 1 June 2018 to 31 May 2019. TTH currently has a bed capacity of 520. The Obstetrics and Gynaecology department has 110 beds and record an average of 600 live births a month. TTH is the only teaching hospital in the northern belt of Ghana and serves as a referral centre for medical/surgical, paediatric and obstetrics, and gynaecological conditions in some parts of the middle belt and the northern belt of Ghana as well as Burkina Faso, which shares a border with the upper regions of Ghana. The study population comprised pregnant women admitted to the in-patient wards of the obstetrics and gynaecology department of TTH during the study period. All pregnant women admitted to the maternity ward with a diagnosis of hypertensive disorders with documented BP measurements were included. Pregnant women with other diagnoses apart from hypertensive disorders were excluded.

Data were collected after reviewing the records of each patient using a structured and pre-tested checklist. All elevated BP categories were matched with the corresponding diagnostic criteria [6] before recording them. Data collected included the age, marital status, educational qualification, NHIS status, employment status, religion, gravida and parity, history of elevated BP in previous pregnancies, antenatal clinic attendance history, hypertensive disorders diagnosed, antihypertensive medicines prescribed and delivery outcomes. Four data collectors and a data quality supervisor were trained before the commencement of data collection. Data were periodically entered into an excel spreadsheet and transferred to SPSS version 22 software for cleaning and analysis [18]. The dependent variable in the study was hypertensive disorders in pregnancy. The independent variables included demographic, obstetric and medical/disease variables. The demographic variables were age, marital status, educational background, employment status and national health insurance scheme (NHIS) status. The obstetric variables were gravida, parity and history of antenatal clinic visits. Medical/disease variables included BP at baseline, past medical history, social history, drug history, duration of pregnancy, previous mode of delivery, BP before delivery and health of the baby and mother. Tables, graphs and pie charts were used to summarise categorical data with frequencies and percentages. The chi-square test was used for tests of association between categorical variables, where a *p*-value less than 0.05 was considered statistically significant. Additionally, the predictors of high blood pressure before delivery were determined using binary logistic regression, where a *p*-value less than 0.05 was considered statistically significant.

## 3. Results

### 3.1. Socio-Demographic Characteristics of Study Participants

Overall, records for 5396 pregnant women were reviewed, of which 673 had hypertensive disorders and, therefore, a prevalence of 12.5%. Table 1 provides a summary of the socio-demographic characteristics of the participants.

### 3.2. Patients’ Gravida and Parity Profile

Seventy-five (11.1%) of the study participants had had no previous pregnancy, 106 (15.8%) had had one pregnancy, 268 (39.8%) had had two or three pregnancies and 224 (33.3%) had had four or more pregnancies. Moreover, 262 (38.9%) had two or three children, 229 (34.0%) had one or no child and 182 (27.0%) had four or more children. Five hundred and sixty-nine (84.6%) respondents had never experienced a miscarriage in pregnancy.

### 3.3. History of Elevated BP in Previous Pregnancies

Out of the 673 pregnant women with elevated BP, 353 (52.4%) had no history of elevated BP in their previous pregnancy and 249 (37.0%) had a history of elevated BP in their previous pregnancy.

### 3.4. Antenatal Clinic History of Study Participants

Three hundred and ninety-five (58.7%) of the participants visited the antenatal clinic (ANC) four or fewer times before delivery, 267 (39.7%) visited the ANC 5–8 times and 14 (2.1%) visited the ANC nine times or more. Furthermore, 387 (57.5%) of the study participants started their ANC visits within 13–26 weeks of pregnancy.

### 3.5. Hypertensive Disorders Diagnosed

Four hundred and sixty-nine (69.7%) pregnant women had complications of elevated BP. These complications were pre-eclampsia, eclampsia and chronic hypertension with superimposed eclampsia. Pre-eclampsia was the highest complication whereas eclampsia was the lowest complication. Figure 1 describes the hypertensive disorders manifested in pregnant women.

### 3.6. Diagnosis of Hypertension during Pregnancy

Forty-seven participants (7.0%) were diagnosed with hypertension before pregnancy, 297 (44.1%), 79 (11.7%) and 250 (37.1%) were within the first, second and third trimesters, respectively. Three hundred and ninety-six (58.8%) of the cases were referred from peripheral health facilities to TTH. Table 2 presents a description of the diagnosis and referrals among study participants.

### 3.7. Antihypertensive Therapy in Pregnancy

The results show that 569 (84.5%) hypertensive pregnant women were on antihypertensive therapy; out of this number, 496 (87.2%) were for a duration of 1–6 months. Four hundred and seventy-three (83.1%) recorded a BP of 140/90 and above during pregnancy; this BP was mostly recorded after 27 weeks of pregnancy. The mean BPs before initiation of antihypertensive therapy, after initiation of antihypertensive therapy and before delivery were 152/97, 126/93 and 122/91, respectively. The most common antihypertensive medication prescribed was sustained-release (SR) oral nifedipine, which was prescribed for 548 (81.4%) participants either alone or with methyldopa, followed by oral methyldopa: 506 (75.2%), intravenous hydralazine: 94 (14.0%), intravenous labetalol: 28 (4.2%) and diuretics: 10 (1.5%). There was a change of medication in 68 (10.1%) participants. These changes were either the removal of methyldopa or nifedipine in those patients who received prescriptions for both due to hypotensive tendencies, or the replacement of both with intravenous hydralazine or labetalol as a result of insignificant BP reduction. With regards to the last BP reading before delivery, 429 (63.7%) had a BP reading between 120/80 and 139/89 with the remaining 244 (36.2%) recording 140/90 and above. Table 3 provides a description of the antihypertensive therapy prescribed for study participants.

### 3.8. Delivery Outcomes of Pregnant Women

In this study, the gestational age at delivery was 27 weeks and above. Three hundred and thirty-seven (96.6%) of the pregnant women whose blood pressures were normal before delivery delivered live babies. The number of dead babies recorded in the study was 38; out of the 38 dead babies, 26 (68.4%) were babies of women with uncontrolled BP after antihypertensive therapy whereas 12 (31.6%) were babies of women with controlled BP. One hundred and ninety-two women with elevated BP before delivery (59.3%) had babies with a birth weight of 2.5 kg or less whereas 112 (27.9%) had babies with Appearance, Pulse Grimace, Activity, Respiration (APGAR) scores of 4.0–6.0 out of 10. There was a statistically significant association between BP control and all delivery outcomes. Table 4 describes the delivery and birth outcomes of the pregnant women and their association with BP control.

### 3.9. Predictors of High BP before Delivery

Results of binary logistic regression analysis to determine the predictors of high BPs of pregnant women before delivery are shown in Table 5. There was a statistically significant relationship between age, educational status and NHIS status, and elevated BP before delivery.

### 3.10. Mode of Delivery

Four hundred and fifty-eight (68.1%) were delivered through caesarean section (CS); out of this number, two hundred and six (45.1%) still had elevated BP after antihypertensive therapy. Two hundred and five (30.5%) of the women delivered per vaginum and 10 (1.5%) through vacuum and forceps.

## 4. Discussion

This study assessed the prevalence, treatment outcomes and pregnancy-related outcomes of pregnant women with hypertensive disorders at TTH in Ghana. In Ghana and Africa at large, many studies have focused on hypertensive disorders without assessing the medications used in their management [3,19,20,21], making this study an important one.

### 4.1. Prevalence of Hypertensive Disorders

The prevalence of hypertensive disorders among the study participants was 12.5%. According to the 2017 annual report of the Obstetrics and Gynecology department of TTH, 439 hypertensive disorders of pregnancy out of a total of 6263 pregnancy-related admissions were recorded. This figure represented 7% of the total admissions for the year and was the highest of all categories of admission [16]. The 5.5% rise in prevalence is an indication that health authorities must have a critical assessment of hypertensive disorders among pregnant women as well as their management in the northern belt of Ghana. This could be achieved through the early detection of risk factors, education on the prevention of modifiable risk factors and the early management of hypertensive disorders in pregnancy. Aside from educating pregnant women, healthcare providers must have continuous professional development to update their knowledge and skills in managing hypertensive disorders in pregnancy. In a systematic review and meta-analysis conducted in Ethiopia, the overall prevalence of hypertensive disorders in pregnancy was 6%, which is about half the prevalence in the current study [22]. The reason for the difference could be the larger sample size in the systematic review. A study from a teaching hospital in Nigeria observed a prevalence of 6.2%, which is also about half of the prevalence in the current study [23]. However, another study conducted by Singh et al., 2014 in a tertiary hospital in Nigeria observed a prevalence of 17% [20]. In a cross-sectional study conducted at the Korle-Bu Teaching Hospital (KBTH) of Ghana between January and February 2013, it was observed that there were a total of 398 women with hypertensive disorders in pregnancy among 1856 deliveries, resulting in a prevalence of 21.4% [24]. The higher prevalence than that observed in the current study could be attributed to the fact that KBTH is a bigger tertiary health facility than TTH, serving higher-risk obstetric emergencies including hypertensive emergencies in pregnancy. Moreover, the differences across studies between and within various countries could be a result of inconsistent terminologies and case definitions in pregnancy-related hypertensive disorders, which impact negatively on the findings from the studies of such disorders. This subject has been discussed extensively by experts in obstetrics and gynaecology leading to the adoption of a consensus statement published by the International Society for the Study of Hypertension in Pregnancy (ISSHP), which is believed to help address the disparities [25].

### 4.2. Socio-Demographic and Clinical Characteristics

Over 70% of the study participants were between the ages of 15 to 35 years implying that the majority of the pregnant women were in their reproductive age. Studies suggest that women generally tend to develop hypertension and diabetes in pregnancy when they conceive after age 35 [26,27]. Therefore, women must be educated to not to give birth after 35 years of age as much as possible whilst reducing the modifiable risk factors of hypertension in pregnancy. This is corroborated by the binary regression analysis in the current study, which showed that there was almost a three-fold likelihood of women aged above 35 years to have elevated BP before delivery compared to those aged 35 years and below. Regarding educational level, the largest group was participants who had received no formal education and accounted for almost two-fifths of the participants. Educated patients tend to understand their disease conditions and adhere to instructions aimed at improving their medical condition from their healthcare providers than uneducated or less-sophisticated patients. This could explain why the binary logistic statistical test showed almost a 10-fold likelihood of women with tertiary education having their BP well controlled compared to uneducated women. This is an indication that efforts must be made to improve female child education, which is neglected in most low-income settings. In the study conducted by Singh et al., 2014 in a Nigerian teaching hospital, 19% had no formal education whereas 81% had formal education [20]. The difference in the educational profile of participants in the Nigerian study and the current study could be due to the difference in the socio-demographic status of the locations. About two-fifths of the study participants had a history of hypertension in their previous pregnancies. One of the risk factors for hypertension in pregnancy is a personal history of hypertension before or during pregnancy [28]. Women with a history of hypertensive disorder in pregnancy must be observed closely and advised appropriately to prevent recurrence. Hypertension before pregnancy was observed in 7% of the pregnant women in the current study. This group of patients tends to develop complications of hypertension in the course of their pregnancy and must be given preconceptual care including counselling and good control of hypertension before the onset of pregnancy. Furthermore, they must be admonished to take their antihypertensive medications during the entire period of pregnancy. The results showed that the most prevalent pregnancy-related hypertensive disorder was pre-eclampsia. Severe forms of pre-eclampsia could cause end-organ damage, cerebral disturbances, pulmonary oedema, impaired liver function and foetal growth restriction, among others [9,29]. These severe forms were shown to be associated with 9% to 26% of global maternal mortality and a significant proportion of preterm delivery, as well as maternal and neonatal morbidity, in a study conducted by Steegers et al. in 2010 [30].

### 4.3. Antihypertensive Therapy in Pregnancy

According to the National Institute of Health Care Excellence (NICE) guidelines, in pregnancy, the target blood pressure should be ≤135/85 mmHg; as such, drug therapy should be aimed at reducing high BPs to that level [31]. The choice of antihypertensive medication during pregnancy is critical in safeguarding the well-being of the unborn baby as well as that of the pregnant woman. Physicians must be abreast with current evidence-based therapies that have been proven to be safe and efficacious through clinical studies. Some antihypertensive classes such as angiotensin-converting enzyme (ACE) inhibitors and angiotensin receptor blockers (ARBs) can restrict foetal growth and even cause foetal deaths; thus, they must be avoided in pregnancy [32,33]. The antihypertensive medicines prescribed in this study were nifedipine, methyldopa, hydralazine and labetalol. Nifedipine and methyldopa were mainly used for the treatment of mild pre-eclampsia, whereas hydralazine and labetalol were used in severe pre-eclampsia and eclampsia. Hydralazine was prescribed for the treatment of severe eclampsia, whereas other forms of hypertension in pregnancy were managed with nifedipine, methyldopa and labetalol. Literature suggests that these aforementioned classes of antihypertensive medications are relatively safe in pregnancy [7,34,35]. The standard treatment guidelines of Ghana also recommend the use of these agents for the various types of hypertensive disorders in pregnancy [36].

### 4.4. Last Blood Pressure Reading before Delivery

The last BP reading before delivery among the pregnant women indicated that 36.2% of them still had elevated blood pressure despite receiving antihypertensive therapy in the course of pregnancy. Elevated blood pressure in pregnancy despite drug treatment is a cause for concern as it could lead to hypertensive emergencies such as stroke and adverse pregnancy outcomes [37]. The poor blood pressure control despite the use of antihypertensive medicines could be attributed to factors such as inadequate doses and an inappropriate combination of antihypertensive medicines, poor adherence to therapy, excessive salt intake and secondary hypertension, among others [38,39,40]. Secondary hypertension, which is highly resistant to antihypertensive medicines, is usually caused by obstructive sleep apnoea, primary aldosteronism and chronic kidney disease [41,42,43]. While secondary hypertension is difficult to manage, factors such as adherence to therapy and reduction in salt intake could be important lifestyle modifications to reduce resistant hypertension. This calls for the education of pregnant women on hypertension prevention during antenatal clinic visits. A drawback to patient education by healthcare providers is illiteracy, which was high in the current study. In such situations, feedback mechanisms could be put in place to enhance the understanding of patients. Non-adherence to antihypertensive medicines usually results from forgetfulness to take the medicines as well as the intention to miss a dose due to a feeling of well-being after taking the medicines for a while [44]. As hypertension is a lifestyle disease that requires lifelong management with medicines and lifestyle modification, patients must be advised as to why they should never miss a dose of their prescription and the need to strictly adhere to the therapy as directed by their physicians.

### 4.5. Mode of Delivery and Delivery Outcomes

Hypertensive disorders in pregnancy tend to cause negative outcomes during delivery. These include induction during labour and prolonging the length of labour, which could cause asphyxia and death of the foetus or newly born baby. Moreover, hypertensive crises during labour could lead to emergency caesarean sections to save both the mother and the unborn baby [45]. Sixty-eight percent of the pregnant women delivered through caesarean sections whereas 30.5% delivered per vaginum. Early or timely delivery is one of the interventions in the management of hypertensive disorders in pregnancy and could be the reason most women had to undergo a caesarean section. However, since the predictors of the birth via vaginum and caesarean section were not ascertained in this study, it will not be possible to determine the exact factors leading to the mode of birth experienced. Future studies could investigate the various predictors using a larger study sample from many healthcare facilities in Ghana. In all, 38 foetal deaths were recorded. Twenty-four of these were babies of mothers with poorly controlled BP. However, these deaths cannot only be attributed to poorly controlled BPs, as other co-morbidities such as maternal age, diabetes mellitus, history of abortion and lack of ANC follow-up, among other factors [12], could have contributed to it. Future studies could focus on the association between these factors and foetal deaths.

### 4.6. Relationship between Blood Pressure Control and Delivery Outcomes

From the results, there was a statistically significant relationship between blood pressure control and delivery outcome. This is an indication that pregnant women diagnosed with hypertension must adhere strictly to their antihypertensive medications as they are important in BP control and pregnancy outcomes. It is beneficial for healthcare providers to devise ways of determining adherence to antihypertensive therapy rather than relying on patient-reported information on adherence, as it is very subjective. A study conducted by Webster et al., 2019 observed that the measurement of urinary antihypertensive metabolites in pregnancy could provide insight into treatment adherence of hypertension in pregnancy [46]. This method, though expensive, provides an objective assessment of adherence.

### 4.7. Study Limitations

The major limitation of this study was the use of secondary data, which are usually fraught with the challenge of missing information. Some important patient information such as maternal weight and height for the calculation of body mass index were missing in most cases and could not be calculated. Further, the lack of inclusion of women with no hypertension during pregnancy in the analysis limits data interpretation.

## 5. Conclusions

The prevalence of hypertensive disorders among pregnant women in the current study was 12.5%. The antihypertensive medicines prescribed (i.e., methyldopa, nifedipine, hydralazine, labetalol and diuretics) were in line with standard treatment guidelines recommended for the management of hypertensive disorders during pregnancy in Ghana. The BPs of about two-thirds of the study participants were well controlled with the antihypertensive therapy. The majority of pregnant women with well-controlled BPs had positive delivery outcomes, whereas most of the negative delivery outcomes were associated with those with uncontrolled BP. It is imperative to periodically educate healthcare teams involved in managing pregnant women in healthcare facilities whilst formulating policies to improve antenatal visits to receive education on lifestyle modification.

## Figures and Tables

**Figure 1 ijerph-20-06153-f001:**
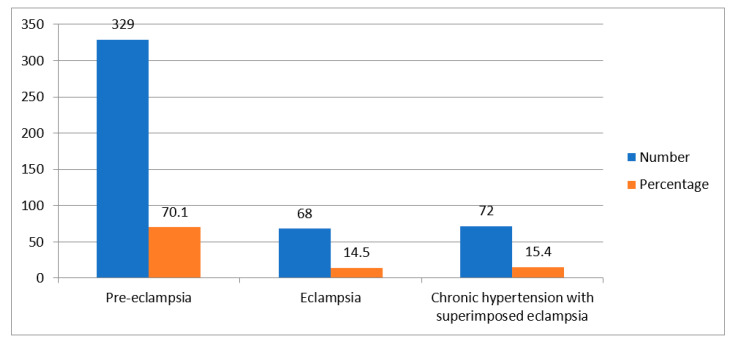
Types of hypertension-related complications.

**Table 1 ijerph-20-06153-t001:** Demographic characteristics of study participants.

Variable		Frequency	Percent
Age in years	15 to 35 years	516	76.7
	≥36 years	157	23.3
	Total	673	100.0
Marital Status	Married	582	86.5
	Single	85	12.6
	Cohabitation	6	0.9
	Total	673	100.0
Educational qualification (highest level)	Nil	267	39.7
	Primary	126	18.7
	JHS/MS *	94	14.0
	SHS	89	13.2
	Tertiary	78	11.6
	Vocational	19	2.8
	Total	673	100.0
Religion	Christian	116	17.2
	Muslim	557	82.8
	Total	673	100.0
Employment status	Employed	430	63.9
	Unemployed	243	36.1
	Total	673	100.0
NHIS status	Insured Non-insured	648 46	96.3 3.7

***** Junior High School (JHS); Middle School (MS); Senior High School (SHS). The Middle School system was replaced by the Junior High School system in September 1987 in Ghana. Vocational Schools are for training individuals who wish to acquire vocations such as catering and carpentry after Junior High School.

**Table 2 ijerph-20-06153-t002:** Stage of pregnancy in which hypertension was diagnosed and referred.

Variable		Frequency	Percent
Period diagnosis of hypertension	Before pregnancy	47	7.0
	1st trimester	297	44.1
	2nd trimester	79	11.7
	3rd trimester	250	37.1
	Total	673	100.0
Referral institution	Maternity home	212	31.5
	Peripheral health facility	178	26.4
	Pharmacy	6	0.9
	Total	396	58.8

**Table 3 ijerph-20-06153-t003:** Antihypertensive therapy and BP measurements in pregnancy.

Variable		Frequency	Percent
Number of participants on antihypertensive therapy		569	84.5
Duration of therapy before delivery	1–6 months	496	87.2
	>6 months	177	12.8
	Total	673	100.0
BP measurement during pregnancy before medication (mmHg)	120/80–139/89	98	14.6
	140/90 and above	575	85.4
	Total	673	100.0
BP measurement 4 weeks after antihypertensive medication (mmHg)	90/60 or less	9	1.3
	120/80–139/89	576	85.6
	140/90 and above	88	13.1
	Total	673	100.0
Changed antihypertensive medication		68	10.1
Patients’ last BP before delivery	120/80–139/89	429	63.7
	140/90 and above	244	36.2
	Total	673	100.0

**Table 4 ijerph-20-06153-t004:** Association between BP control and delivery outcomes.

Variable	Normal BP No. (%)	Elevated BP No. (%)	*p* Values
Delivery			
Live baby	337 (96.6)	298 (92.0)	0.009
Dead baby	12 (31.6)	26 (68.4)	
Birth weight (kg)			
2.5 or less	64 (18.3)	192 (59.3)	0.007
2.6–3.5	256 (73.3)	118 (36.4)	
≥3.6	29 (8.3)	14 (4.3)	
APGAR score			
4.0–6.0	55 (23.5)	112 (27.9)	0.001
≥7.0	179 (76.5)	289 (72.1)	

**Table 5 ijerph-20-06153-t005:** Summary of binary logistic regression for high BP before delivery versus patient characteristics.

Variable	Wald	OR (95% CI)	*p*-Value
Age			
15–35	Reference		
≥36	3.93	2.63 (2.38–3.12)	0.004
Marital status			
Married	Reference		
Single	2.68	1.19 (0.86–1.52)	0.237
Cohabitation	1.74	1.32 (1.02–1.53)	0.094
Educational Qualification			
Nil	Reference		
Primary	3.95	1.60 (1.38–2.24)	0.016
JHS/MSLC	6.43	2.03 (1.77–2.79)	0.037
SHS	4.17	3.77 (3.39–4.14)	0.008
Tertiary	4.62	9.69 (9.41–10.26)	0.029
Vocational	6.33	3.26 (2.87–3.62)	0.001
Employment status			
Employed	Reference		
Unemployed	2.82	0.48 (0.12–0.76)	0.649
NHIS status			
Insured	Reference		
Non-insured	3.51	0.61 (0.23–1.02)	0.001

## Data Availability

Not applicable.

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
