# Peer review of "Prevalence of Hypertensive Disorders, Antihypertensive Therapy and Pregnancy Outcomes among Pregnant Women: A Retrospective Review of Cases at Tamale Teaching Hospital, Ghana"

_ijerph, 2023, doi:10.3390/ijerph20126153_

Round 1
Reviewer 1 Report
The authors describe hypertensive disorders of pregnancy as the leading cause of death. It would be very good if they talked about or described at least the 3 main causes of death in Ghana and what about disorders such as hemorrhage and sepsis?
It is important to emphasize that maternal death in most cases is preventable, so not only is it poor medical care, but the first contact team is not sufficiently trained to diagnose those women at risk early . I mean, education is probably needed.
Agree that women with hypertension should attend the doctor for surveillance as well as those who have risk factors; however, it is important to consider that for some of them it is not so easy, since they have to travel great distances to have access to health and even more so in countries with medium and/or low resources.
I consider it important to emphasize that the study does not evaluate the antihypertensive drugs used, but that blood pressure control is associated with better perinatal outcomes.
I believe that in countries where maternal death is high, the authorities should pay attention to the training of health personnel, favoring public policies that support education.
In the paragraph where they state that women must be educated to go to the doctor, it is important to highlight that not only women must be educated, but also the medical team that is in charge of caring for them.
Regarding the genesis of hypertensive disorders of pregnancy, there seems to be little evidence regarding lifestyle; in most cases it is rather placental insufficiency that causes them.
I think that as part of the conclusions it is important to insist that more than the choice of antihypertensive it is the control of blood pressure that changes the natural history of perinatal outcomes.
In the conclusions, I suggest that the importance of educating the health team and the change in public policies be included.
Reviewer 2 Report
The presented research results are not very innovative, because they present a well-known problem of complications in non-compliance patients. In this case, it is important to know that this applies to a large group of pregnant women and their offspring. The results should include information on the dosage of antihypertensive drugs used, other pharmacotherapy, pregnant women's body weight (BMI) or other comorbidities, e.g. diabetes, thyroid dysfunction and others. Most of the references are older than from the last 5 years. The form used to catalog the clinical data was not provided.
Reviewer 3 Report
The manuscript describes a 1-year retrospective study undertaken at a major teaching hospital in Ghana in women with hypertensive disorders of pregnancy. Prevalence of this disorder was described as well as the relationship between multiple patient characteristics an the risk for hypertensive disorders of pregnancy and, the relationship with hypertensive disorders of pregnancy and fetal death was described. This is an interesting manuscript. However, there are some substantive concerns:
1) In the abstract, please provide the percent of babies that died after being born to mothers without any hypertensive disorder of pregnancy - having an event rate contrast among births to unaffected women provides an important reference point
2) there is no mention of maternal mortality - this should be included
3) In the methods and materials please provide the explicit case-definition for the respective hypertensive disorders of pregnancy - one consideration is to simply describe, for example, pre-eclampsia and not the subtypes of pre-eclampsia as has currently been done; these sub-types are mentioned but never defined. A suggestion, this level of detail adds nothing to the paper
4) please provide an explicit description of how women with hypertensive disorders of pregnancy included in this analysis were identified - were ICD codes used??
5) in the results there is no need to repeat verbatim in the text numbers that are cited in tables - the first paragraph of the results is one example of this redundancy
6) in table 3, please additionally provide average or median BPs for the parts of the table where BP has been categorized
7) The data from table 4 should be expressed as a percent contrast of women who delivered dead babies with and without any hypertensive disorder of pregnancy
8) the level of detail in figure 1 is excessive - there is no need to break down preeclampsia into severe, mild and eminent nor is there a need to break down eclampsia into a category of severe as none of these stratifications has been defined and their inclusion adds nothing to the paper
9) on page 5, at the bottom of the page, describe the intensity of drug therapy in medically treated with hypertensive disorders of pregnancy minimally by giving the frequency of those treated with 1, 2, 3 or 4 or more drugs
10) I am assuming that the prevalence of IVF is very low. However, if not, does this history impact risk for developing a hypertensive disorder of pregnancy?
11) the discussion is too long and should be shortened minimally by one-third
MINOR
1) in table 5 there are definitions use that have not been defined - JHS, MSLC, SHS; also, if terms like vocational and primary are deemed important enough to use in these tables then either define in the methods or minimally define them in a table footnote
2) consider breaking up the discussion into more paragraphs
3) clarify if the use of nifedipine was nifedipine ER or SR not short-acting nifedipine as the latter was never approved, at least in the USA, for hypertension treatment
4) consider discussing what BP thresholds should be used for initiating antihypertensive drug therapy in pregnant women and also what the target BP should be
Round 2
Reviewer 3 Report
The manuscript has chosen to depict most of the results only in women with hypertensive disorders in pregnancy making it hard to really understand the incremental risk in this group compared to women who had not hypertension while pregnant. For example, the data regarding fetal death and BP control has been restricted to women with hypertensive disorders of pregnancy whereas a more traditional approach would also display the deaths in women without hypertensive disorders allowing the reader to understand the incremental risk according to the reference group not just the risk inside of hypertensive group according to level of BP control.
It is again stated that women without hypertensive disorders were excluded. That is not true because to calculate prevalence estimates you had to include them in the assembled data. Please correct this statement in the methods.
The lack of inclusion of women with no hypertension during pregnancy in the analysis limits data interpretation and should be included as a limitation.
